# Non-Pathological Psychological Distress among Mainland Chinese in Canada and Its Sociodemographic Risk Factors amidst the Pandemic

**DOI:** 10.3390/healthcare10112326

**Published:** 2022-11-21

**Authors:** Lixia Yang, Linke Yu, Kesaan Kandasamy, Yiran Wang, Fuyan Shi, Weiguo Zhang, Peizhong Peter Wang

**Affiliations:** 1Department of Psychology, Toronto Metropolitan University, Toronto, ON M5B 2K3, Canada; 2The Centre for New Immigrant Well-Being (CNIW), Markham, ON L3R 6G2, Canada; 3Dalla Lana School of Public Health, University of Toronto, Toronto, ON M5T 3M7, Canada; 4School of Public Health, Weifang Medical University, Weifang 261053, China; 5Department of Sociology, University of Toronto Mississauga, Mississauga, ON L5L 1C6, Canada; 6Division of Community Health and Humanities, Faculty of Medicine, Memorial University of Newfoundland, St. John’s, NL A1C 5S7, Canada

**Keywords:** COVID-19, psychological distress, Mainland Chinese in Canada, health disparity, sociodemographic predictors

## Abstract

The COVID-19 pandemic has exacerbated health inequalities, with a potentially heightened mental health risk for Mainland Chinese in Canada, given the rising anti-Chinese discrimination, and barriers in assessing health services. In this context, this study aimed to assess non-pathological psychological distress towards COVID-19 and identify its sociodemographic risk factors among Mainland Chinese in Canada at the early stages of the pandemic. Methods: A sample of 731 Mainland Chinese aged 16 or older completed an on-line survey to examine their attitudes, behavioural, and psychological responses towards COVID-19. Non-pathological psychological distress was assessed with a 7-item self-report scale to capture common emotional reactions towards COVID-19. Results: A factor analysis revealed a single-factor structure of the 7-item COVID-19 psychological distress scale (Eigen λ = 3.79). A composite psychological distress index (PDI) score was calculated from these items and used as the outcome variable. Multivariate regression models identified age, financial satisfaction, health status, and perceived/experienced discrimination as significant predictors of psychological distress (*p*s ≤ 0.05). Conclusions: Mainland Chinese in Canada who were over 25, in poor financial/health status, or with perceived/experienced discrimination were at a higher risk for COVID-19-related psychological distress. The health inequity across these factors would inform the services to mitigate mental health risk in minority groups.

## 1. Introduction

The coronavirus (COVID-19) pandemic has been spreading widely in Canada [1]. It has exacerbated health inequalities that have long persisted in pre-pandemic Canada, with racialized people, women, unemployed individuals, homeless individuals, or those with low incomes being disproportionately more affected by the pandemic [2]. In this context, there is a growing interest in assessing the psychological impact of the pandemic across the globe [3,4,5,6]. Accumulated research suggests that mental health has been negatively impacted in Canada, considering the strict quarantine and social-distancing measures to control the spread of the virus at the early stages of the pandemic [7,8]. Furthermore, vulnerable populations such as individuals with disabilities [9], immigrants, and indigenous peoples reported especially poor mental health during the pandemic [10].

It is reasonable to expect that the mental health risk will be heightened in visible minority populations, especially among Chinese migrants, considering their experiences with cultural, language, and social barriers in accessing healthcare services [11,12,13]. Furthermore, with a close tie to China, Mainland Chinese abroad are targets of racism and discrimination during the pandemic. Past research has demonstrated that Chinese individuals worldwide react to such racism and xenophobia with frustration [14] and this may further jeopardize their emotional health. In fact, past research has identified perceived discrimination as a significant mental health risk predictor in immigrant and refugee populations [15], including Chinese residents in Canada during the pandemic [16,17].

### 1.1. Psychosocial Impact of COVID-19

The COVID-19 outbreak has a detrimental impact on psychological wellbeing across the world [5,7]. A systematic review suggests that social isolation during quarantine strongly correlated with mental health symptoms [18]. The impact may span multiple domains. For example, health workers during the severe acute respiratory syndrome (SARS) outbreak commonly endorsed fear of social rejection and stigmatization, along with several other psychological symptoms, such as anxiety, avoidance, uncertainty, and distress-related somatic problems [19]. Symptoms of mood disorders also appeared to be prevalent during the COVID-19 pandemic. A national survey at the early stages of the pandemic suggested that Canadians reported a twofold increase in severe depressive symptoms and nearly a fourfold increase in severe anxiety symptoms [8]. Surveys in Mainland China revealed that roughly a third of respondents experienced anxiety or depression symptoms during the COVID-19 pandemic [20,21]. Overall, the pandemic outbreak did cause concerning mental health impacts.

### 1.2. Vulnerability of Mainland Chinese in Canada

In the process of adapting to Western life, Mainland Chinese in Canada may face numerous obstacles such as social, cultural, and language barriers, as well as racial discrimination [22]. These obstacles may place them at a disadvantage in achieving personally meaningful goals, which may further negatively impact their psychological wellbeing [23]. Considering the double jeopardy hypothesis [24], Mainland Chinese in Canada as a racialized minority group might be especially vulnerable to the mental health impacts of the pandemic. Past research showed that Chinese in Canada might be particularly vulnerable to psychological distress, given their likelihood to experience stigma related to using mental health services [22,25,26], and their preference for traditional Chinese health practices [27]. Canadian survey data suggest that immigrants’ mental health condition has drastically declined following the COVID-19 outbreak and immigrants tend to endorse a higher number of generalized anxiety disorder symptoms than Canadian-born responders [28]. Moreover, the detrimental mental health impact of the COVID-19 pandemic tends to be more prolonged in Chinese immigrants, as demonstrated in a higher rate of post-traumatic stress symptoms in Mainland Chinese than their Spanish and American counterparts [29,30]. Taken together, the mental health impacts of the pandemic are particularly concerning among Chinese in Canada. Thus, it is very important to assess the psychological distress of this vulnerable population.

### 1.3. Sociodemographic Variables

A few sociodemographic variables have been identified as being linked to psychological distress during the pandemic. For example, young (aged 18 to 30 years old) and older adults (aged 65 years old and over) showed significantly higher levels of psychological distress relative to other age groups [21]. Women tended to report more psychological distress symptoms than men in Mainland China [21,31]. Unsurprisingly, poor health status predicted greater psychological distress [16,31]. Similarly, in Canadian samples, those who were at higher risk for losing their jobs and those with financial insecurity reported a higher level of worry [32]. Considering that individuals with lower education or those who were currently unemployed were less likely to use mental health services [33], it is not a surprise that individuals with lower education also reported greater psychological distress during the pandemic [21]. Immigrants were at an increased risk for discrimination, which was linked to poor mental health outcomes [15]. Recent poll data showed that nearly two thirds of newly-migrated Canadians (64%) were worried about the rising discrimination during the pandemic [34]. Taken together, past research has identified age, gender, education, and health/income/employment condition as risk factors for psychological distress during the pandemic. Discrimination was also identified as a mental health risk factor in immigrant population [35]. These results set a foundation for the current study, which aimed to identify sociodemographic risk factors for non-pathological psychological distress towards the pandemic among Mainland Chinese in Canada, a multicultural society.

### 1.4. The Aim of the Current Study

Despite the possibly heightened risk of psychological distress in Mainland Chinese, there is a scarcity in Canadian research on the psychological impact of COVID-19 among this minority population, particularly at the initial outbreak stage of the pandemic. As a rapid response to the pandemic outbreak, the current study aimed to fill this gap to have a brief assessment of non-pathological psychological distress towards COVID-19 pandemic and identify related sociodemographic predictors among Mainland Chinese in Canada at the early stages of the pandemic. This is important given the time-sensitive behavior measures implemented at that time. The results will shed light on future public health policies and measures to best mitigate psychological distress or mental health issues at the initial stages of the pandemic outbreak.

Therefore, this study addressed a specific and explorative research question: whether higher psychological distress of Mainland Chinese in Canada at the initial outbreak stage of the pandemic was predicated by their sociodemographic profiles (e.g., age, gender, education, financial/health condition, employment status, and discrimination)? In light of the previous findings reported in the literature [16,21,31,32,33], it was hypothesized that higher psychological distress would be predicted by sociodemographic variables such as younger age, being a woman, poor health/financial status, lower education, unstable employment, and perceived or experienced anti-Chinese discrimination.

## 2. Materials and Methods

### 2.1. Participants

Participants were primarily recruited through a snowball sampling approach via WeChat (a most popular social media platform among Chinese), including private and group chat forums and moment posts, as well as other online platforms such as emails (to the collaborative Chinese community organizations) and websites, participating the lab contacts and the website of The Centre for New Immigrant Well-being (CNIW) in parallel. A total of 764 individuals responded to the survey, in which 731 completed the informed consent and met the inclusion criterion: Mainland Chinese aged 16 or older who had lived in or would live in Canada for at least 6 months. Compared to the 2016 Canadian census results on Mainland Chinese immigrants (55% female, 70% aged 25–64) [36], women (66.21%) and those aged 25–64 (84.13%) were slightly overrepresented in our sample. A majority of the sample were Ontario residents (83.99%), married or with common-law relationship (75.65%), with a university degree or higher (77.70%), and not involved in healthcare work (92.74%). About 22.80% of the sample reported either confirmed or uncertain contact with someone who had flu/cold-like symptoms, approximately 16.71% indicated that they experienced such symptoms, and 26.85% reported (10.40%) or unsure (16.42%) about whether they experienced discrimination related to COVID-19. Table 1 presents the sample characteristics.

### 2.2. The Survey

A cross-sectional online survey was conducted in Mandarin at the early stage of the pandemic (i.e., 2 April to 13 May 2020). Participation was anonymous and voluntary, and informed consent was obtained before participation. No incentive was provided for participation. Identifying information (e.g., WeChat ID, IP address) was removed before analysis. The survey is composed of two sections. The first section gathers general sociodemographic information about participants and the second section collects information on psychological, cognitive, and behavioral responses towards COVID-19. In light of previous findings [2], we included questions on sociodemographic information (e.g., gender, employment status, income and financial satisfaction, and living conditions) to have a comprehensive understanding of risk factors in this vulnerable minoritized group. The financial satisfaction, health status, and perceived discrimination were rated with a 5-point Likert Scale: from 1 (very dissatisfied) to 5 (very satisfied) for financial satisfaction, from 1 (poor) to 5 (excellent) for health status, and from 1 (strongly disagree) to 5 (strongly agree) for perceived discrimination. Table 1 lists these sociodemographic variables and the corresponding response options for each of them.

### 2.3. The COVID-19 Psychological Distress Scale

The COVID-19-induced non-pathological psychological distress was assessed with a 7-item self-report scale that captures commonly reported emotions (uneasy, fearful, anxious, depressed, stressed, indecisive, and confused) in response to COVID-19. The items read as “please rate to which degree you feel at ease/fearful/anxious/depressed/stressed/indecisive/confused about the current COVID-19 pandemic?” respectively for each emotional response. Participants rated how each item described themselves based on a 5-point Likert Scale from 1 (not at all) to 5 (very much). To minimize habitual or robot responses, one item was worded in a positive valence (i.e., at ease) and, thus the score was reverse coded to index “uneasy”.

The rating scores showed a high reliability (Cronbach’s α = 0.85) across the seven items (*n* = 481), with listwise deletion of missing value cases. The Pearson inter-item correlation analysis showed consistent positive correlations among all these items (*r*s = 0.24–0.68, *p* < 0.001). The principal component factor analysis extracted a single-factor component (*λ* = 3.79, accounting for 54.13% variance). The seven items heavily loaded on this component (0.53–0.84). Thus, a composite psychological distress index (PDI) score was calculated as the average across the seven items, with missing values being replaced with the average score for each participant. Furthermore, the PDI level was defined by categorizing scores larger than the scale median point (i.e., 3) as “high-distress” (50.48%), and the remaining as “low-distress”.

### 2.4. Data Analysis

The data analysis was performed in IBM SPSS 24. A descriptive analysis was conducted to show the sociodemographic characteristics for the whole sample and samples stratified by the Psychological Distress Index (PDI) level category (see Table 1). Group differences in PDI level were analyzed with the Pearson’s chi-square test (Table 1). The effect of each sociodemographic variable on the PDI score was assessed with a T-test or One-way Analysis of Variance (ANOVA) model (Table 2). The significant sociodemographic risk predictors were further identified with a multivariate linear regression model (with PDI as the dependent variable) and a multivariate logistic regression model (with the PDI level as the dependent variable).

## 3. Results

### 3.1. The Relationship between the Sociodemographic Variables and PDI

Pearson Chi-square analyses (Table 1) showed that distress level classification (i.e., “high-distress” vs. “low-distress”) was related to age, financial satisfaction, health status, perceived and experienced discrimination, contact with others showing flu/cold-like symptoms, and experiences of flu-like symptoms (*χ^2^*s ≥ 3.99, *p*s < 0.05). Young adults (aged < 25), those who were financially satisfied, healthy, with little to no perceived or experienced discrimination, or without exposure or experience of the symptoms, were less likely to be in the “high-distress” category. Additionally, the independent samples t-tests or one-way ANOVA (Table 2) on the continuous PDI score showed significant effects of age group, gender, financial satisfaction, health status, perceived and experienced discrimination on the psychological distress score (*p*s ≤ 0.017). Specifically, young adults reported less distress than middle-aged adults. Women reported higher distress than men. Those who were financially dissatisfied, in poor health, and with perceived or experienced anti-Chinese discrimination reported differentially higher distress. Taken together, the Chi-square and group difference tests suggested a higher psychological distress reported by women, those aged over 25, financially dissatisfied, in poor health, with perceived and experienced discrimination, or experiencing flu-like symptoms.

### 3.2. Regression: The Sociodemographic Prediction for PDI and PDI Level

A multivariate linear regression model was conducted on the continuous PDI score and a multivariate logistic regression model was conducted on the categorial PDI level score to identify significant sociodemographic predictors for PDI (see Table 3). All the sociodemographic variables that showed a significant relationship with PDI (or PDI level) in the aforementioned Chi-square analyses or t-tests/ANOVAs were entered as predictors in these models, including age group, gender, financial satisfaction, health status, perceived and experienced discrimination, contact with others showing flu-like symptoms, or experience flu-like symptoms. The results identified age group, financial satisfaction, health status, perceived/experienced discrimination as significant predictors for psychological distress. Specifically, young adults aged below 25 reported to be less distressed than their middle-aged or older counterparts. Individuals who were financially satisfied, physically healthy, or with little-to-no perceived/experienced discrimination reported to be less distressed (Table 3).

## 4. Discussion

This study addressed an explorative research question to assess non-pathological psychological distress towards COVID−19 among Mainland Chinese in Canada and identify its sociodemographic predictors at the initial outbreak stage of the pandemic. The results showed that distress level is lower for young adults (aged < 25), and higher for those in poor financial/health condition or who perceived or experienced discrimination. These results are discussed below in light of the literature.

### 4.1. Sociodemographic Risk Factors for Psychological Distress

The finding of lower psychological distress risk in young Chinese relative to other age groups was not fully consistent with the literature. It has been reported that both young adults (aged 18–30) and the elderly (aged above 60) in Mainland China reported the highest level of psychological distress related to COVID−19 [18]. It has also been reported that the mental health of young or middle-aged adults, and those with disabilities was more impacted by the pandemic [9,37,38]. But other studies also reported that older adults experienced enhanced health anxiety and emotional loneliness following the implementation of quarantine measures [39,40]. Our data identified young adults as being least vulnerable to psychological distress, but probably for different reasons. Middle-aged Chinese in Canada typically serve as the foundational support (financial or social) for families and communities [41]. They have family commitments (e.g., raising children, supporting old parents) and need to fulfill social duties (e.g., jobs or community services). Thus, they are most likely to experience exacerbated family/work-related stress during the pandemic, which threatens family harmony, work security, as well as work-life balance. Due to cultural and language barriers, older adults (80.95% retired) in the sample might experience heightened stress levels related to poor health, social isolation and elder abuse. These stressors, however, are less likely to be applicable to the youth (aged 18–25) as most of them are international students or young immigrants (81.13%) who might still be dependent on their parents for financial support, and not yet bound by caregiving responsibilities. Nevertheless, this novel finding is worth further investigation.

Our results suggest that individuals who are unsatisfied with their financial or health conditions reported higher level of psychological distress, a result consistent with the commonly reported associations between socioeconomic status (SES) and mental health [42,43]. The results are generally consistent with the double jeopardy hypothesis [24], suggesting that financially-related or health-stratified health inequity is particularly enhanced in visible minority groups [23,44], particularly for Chinese, who were likely to experience increased incidences of COVID-19-related discrimination [45]. Consistently, our results also identified both perceived and experienced discrimination, especially perceived discrimination, as significant predictors for non-pathological psychological distress among Mainland Chinese in Canada. This result identifies discrimination as an important mental health disparity risk predictor for racialized Mainland Chinese in Canada during the pandemic. Somewhat consistent with this result, it has been found that discrimination experience was associated with both physical and psychological distress [46] and racialized groups in Canada suffered a higher incidence and more severe consequences from COVID-19 derived health risks [47]. However, it should be noted that the incidence of personal experience of discrimination (10.4%) in our study was lower than the incidence reported in a survey (50%) conducted by the Angus Reid Institute (ARI) [48]. This difference may be due to cultural deviations in discrimination definition, considering that all of the respondents in our survey were born in Mainland China relative to the sample in the ARI survey (only 22%). Respondents born in different countries/areas may hold different understanding and criteria for discrimination, given the enhanced health disparity experienced by visible minority groups in Canada. Additionally, the restriction of outdoor activities as widely reported in our sample also minimized the chance to encounter discrimination.

Consistent with the available COVID-19 literature [31], other sociodemographic variables, such as living arrangements, were not significant predictors for psychological distress. It is interesting that education status did not predict psychological distress, but this may be due to limited variability, given that 77.70% of the sample completed at least University-level education. Notably, gender did not significantly predict psychological distress. Despite the findings that women are more likely to develop major depressive disorder, recent meta-analytic suggests that gender differences are much smaller when examining depressive symptoms [49]. Taken together, the results from this study clearly depict mental health disparities by identifying some risk factors (e.g., age, financial/health status, and discrimination) for psychological distress experienced by Mainland Chinese in Canada, a racialized minority group.

In light of the research question of this study to assess the non-pathological psychological distress towards COVID-19 and identify its sociodemographic predictors among Mainland Chinese in Canada, we found that over half of the sample scored in the “high-distress” category (50.48%) at the early pandemic outbreak stage. Furthermore, the results identified age, financial/health status, and discrimination as significant risk predictors for psychological distress in this population. Specifically, the distress level was lower for young adults (aged < 25), and higher for those in poor financial/health condition or those who perceived or experienced anti-Chinese discrimination.

### 4.2. Limitations

Cultural integration, conceptualized as the adoption of the host and the origin cultures, has been linked with better subjective wellbeing compared to immigrants who have assimilated entirely to the host culture [50]. The lack of such an assessment in the current study impeded the capacity for evaluating the association between psychological distress and acculturation. Another limitation was the representativeness of the sample, which was partially related to our recruitment methods that relied on using a social media platform (i.e., WeChat). The sample was also largely comprised of women, and highly educated Chinese living in Ontario, which limits the generalizability of our study results. This, combined with a small sample size, may limit sample variability in certain categories (e.g., medical workers), restricting the sample representativeness and statistical power. Finally, the current study also employed a single 7-item psychological distress scale created by the research team. Further studies are needed to validate this scale with standardized psychological distress or mental health measures.

### 4.3. Conclusions

Nonetheless, this study provided critical information on psychological responses to the initial pandemic outbreak among Mainland Chinese in Canada. The results generally identified older age (middle-aged and older adults relative to young adults), poor financial and health status, and perceived anti-Chinese discrimination as significant risk factors for higher psychological distress among this vulnerable population.

Given the well-reported health inequalities among racialized individuals in Canada [2], and the dramatically rising cases of racism and discrimination against Chinese (e.g., a 30% increase in harassment or attacks against Chinese Canadians in 2020 since the start of the COVID-19 outbreak) [51], as well as the more prolonged mental health impacts of the pandemic on this population [29], Chinese Canadians are under a growing concern for mental health issues during and following the pandemic. However, they are also underserved in mental health support and services due to cultural, language, and social barriers in accessing healthcare resources [11,12,13]. Future research may follow up and further address the mental health condition of this population and associated risk factors in greater depth and with standardized measures. By identifying sociodemographic risk predictors for psychological distress at the initial stage of the COVID-19 outbreak, we could quickly identify and target high-risk individuals for psychological prevention and intervention services. Thus, these results will inform future public health policy and measures to best mitigate non-pathological psychological distress at the initial pandemic outbreak stage in the future. The findings may guide public health policy decisions and intervention/prevention program development related to mental health services to address and minimize the heightened health disparity in this vulnerable but under-served population.

## Figures and Tables

**Table 1 healthcare-10-02326-t001:** Sample Characteristics.

Variables		All	Low-Distress ^a^	High-Distress ^a^	*χ^2^*
		*N* (%)	*N* (%)	*N* (%)	
Age group	<25	53 (7.25)	40 (11.05)	13 (3.52)	16.559 ***
	25–64	615 (84.13)	288 (79.56)	327 (88.62)	
	≥65	63 (8.62)	34 (9.39)	29 (7.86)	
Sex	Male	247 (33.79)	133 (36.74)	114 (30.89)	2.791
	Female	484 (66.21)	229 (63.26)	255 (69.11)	
Marital status	Married /Partnered	553 (75.65)	277 (76.51)	276 (74.80)	0.294
	Other	178 (24.35)	85 (23.48)	93 (25.20)	
Highest degree	University or higher	568 (77.70)	286 (79.01)	282 (76.42)	0.704
	Other	163 (22.30)	76 (20.99)	87 (23.58)	
Length in Canada	≤5 years	162 (22.16)	77 (21.27)	85 (23.04)	0.330
	5 years	569 (77.84)	285 (78.73)	284 (76.96)	
Living arrangement	Alone	76 (10.43)	40 (11.05)	36 (9.81)	0.300
	Others	653 (89.57)	322 (88.95)	331 (90.19)	
Employment	Student	68 (9.30)	38 (10.50)	30 (8.13)	1.496
	Employed /Self-employed	423 (57.87)	210 (58.01)	213 (57.72)	
	Other	240 (32.83)	114 (31.49)	126 (34.15)	
Healthcare worker	Yes	52 (7.11)	22 (6.08)	30 (8.13)	1.187
	No	678 (92.74)	340 (93.92)	338 (91.60)	
Financial satisfaction	Dissatisfied	136 (19.24)	40 (11.27)	96 (27.27)	36.519 ***
	Neutral	285 (40.31)	141 (39.72)	144 (40.91)	
	Satisfied	286 (40.45)	174 (49.01)	112 (31.82)	
Health status	Dissatisfied	31 (4.31)	7 (1.96)	24 (6.63)	10.080 **
	Neutral	187 (25.97)	91 (25.42)	96 (26.52)	
	Satisfied	502 (69.72)	260 (72.63)	242 (66.85)	
Perceived discrimination	Disagree	258 (41.68)	146 (51.77)	112 (33.23)	31.553 ***
	Neutral	197 (31.83)	89 (31.56)	108 (32.05)	
	Agree	164 (26.49)	47 (16.67)	117 (34.72)	
Experienced discrimination	Yes/Unsure	196 (26.85)	70 (19.39)	126 (34.15)	20.229 ***
	No	534 (73.15)	291 (80.61)	243 (65.85)	
Contact others with symptoms	Yes/Unsure	166 (22.80)	71 (19.67)	95 (25.89)	3.997 *
	No	562 (77.20)	290 (80.33)	272 (74.11)	
Experience of symptoms	Yes/Unsure	122 (16.71)	50 (13.81)	72 (19.57)	4.339 *
	No	608 (83.29)	312 (86.19)	296 (80.43)	

Note. ^a^ Low-distress (PDI ≤ 3) and High-distress (PDI > 3). PDI = Psychological Distress Index; * *p* < 0.05, ** *p* < 0.01, *** *p* < 0.001.

**Table 2 healthcare-10-02326-t002:** Independent sample difference test on PDI stratified by sociodemographic variables.

Variables	Level (Code)	PDI	T-Test/ANOVA (*F/t* Value)	Multiple Comparison ^a^	
		*M (SD)*		Comparison	*p*
Age group	<25 (1)	2.81 (0.73)	5.09 **	1 vs. 2	0.004 **
	25–64 (2)	3.18 (0.81)		1 vs. 3	0.064
	≥65 (3)	3.16 (0.92)		2 vs. 3	1.00
Sex	Male (1)	3.05 (0.79)	−2.40 *		
	Female (2)	3.21 (0.83)			
Marital status	Married /Partnered (2)	3.14 (0.81)	0.71		
	Other (1)	3.19 (0.84)			
Highest degree	University or higher (2)	3.14 (0.81)	0.82		
	Other (1)	3.20 (0.83)			
Length in Canada	≤5 years (1)	3.14 (0.84)	−0.20		
	>5 years (2)	3.16 (0.81)			
Living arrangement	Alone (1)	3.09 (0.83)	−0.75		
	Others (2)	3.16 (0.82)			
Employment	Student (1)	3.06 (0.90)	0.59		
	Employed /Self-employed (2)	3.15 (0.78)			
	Other (3)	3.19 (0.85)			
Healthcare worker	Yes (1)	3.34 (0.95)	1.66		
	No (2)	3.14 (0.80)			
Financial satisfaction	Dissatisfied (1)	3.54 (0.85)	28.73 ***	1 vs. 2	<0.001
	Neutral (2)	3.17 (0.77)		1 vs. 3	<0.001
	Satisfied (3)	2.93 (0.75)		2 vs. 3	0.001
Health status	Dissatisfied (1)	3.62 (0.83)	7.61 **	1 vs. 2	0.042
	Neutral (2)	3.23 (0.82)		1 vs. 3	0.001
	Satisfied (3)	3.09 (0.80)		2 vs. 3	0.118
Perceived discrimination	Disagree (1)	3.01 (0.77)	26.51 ***	1 vs. 2	0.043
	Neutral (2)	3.19 (0.76)		1 vs. 3	<0.001
	Agree (3)	3.58 (0.85)		2 vs. 3	<0.001
Experienced discrimination	Yes/Unsure (1)	3.38 (0.79)	4.62 **		
	No (2)	3.07 (0.81)			
Contact others with symptoms	Yes/Unsure (1)	3.24 (0.76)	1.50		
	No (2)	3.13 (0.83)			
Experience of symptoms	Yes/Unsure (1)	3.34 (0.71)	2.70		
	No (2)	3.12 (0.83)			

Note. ^a^
*p* values for the multiple comparisons (based on the code of levels for each variable) with Bonferroni correction. *M* = Mean, *SD* = Standard Deviation. * *p* < 0.05, ** *p* < 0.01; *** *p* < 0.001.

**Table 3 healthcare-10-02326-t003:** Regression models on the Sociodemographic Predictors for PDI score and PDI level (“high distress” vs. “low distress”).

Predictors		Linear Regression on PDI Score		Logistic Regression on PDI Level	
		*β*	95% *CI*	*OR*	95% *CI*
Age Group	<25 (reference)				
	25–64	0.37 **	0.14, 0.61	4.50 ***	2.13, 9.51
	≥65	0.40 *	0.07, 0.73	3.40 *	1.27, 9.10
Gender	Male (reference)				
	Female	0.08	−0.06, 0.21	1.01	0.69, 1.47
Financial satisfaction	Dissatisfied (reference)				
	Neutral	−0.26 **	−0.42, −0.09	0.49 **	0.30, 0.82
	Satisfied	−0.50 ***	−0.68, −0.33	0.30 ***	0.18, 0.50
Health satisfaction	Dissatisfied (reference)				
	Neutral	−0.22	−0.53, 0.09	0.39	0.14, 1.09
	Satisfied	−0.31 *	−0.61, −0.01	0.43	0.16, 1.15
Perceived discrimination	Disagree (reference)				
	Neutral	0.10	−0.04, 0.25	1.33	0.88, 1.99
	Agree	0.44 ***	0.28, 0.60	2.63 ***	1.64, 4.21
Experienced discrimination	Yes/Unsure (reference)				
	No	−0.14 ^ǂ^	−0.29, 0.001	0.67	0.44, 1.01
Contact others with symptoms	Yes/Unsure (reference)				
	No	0.07	−0.09, 0.23	0.96	0.60, 1.51
Experience of symptoms	Yes/Unsure (reference)				
	No	−0.13	−0.31, 0.04	0.87	0.52, 1.43
Constant		3.52	2.94, 4.09	1.63	-

Note. PDI = Psychological Distress Index, OR = Odds Ratio, CI = Confidence of Interval. ^ǂ^
*p* = 0.05, * *p* < 0.05, ** *p* < 0.01; *** *p* < 0.001.

## Data Availability

The data files, the SPSS syntax file and the variable label index file could be retrieved from osf.io/4vr6y (accessed on 19 November 2022).

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
