# Peer review of "Non-Pathological Psychological Distress among Mainland Chinese in Canada and Its Sociodemographic Risk Factors amidst the Pandemic"

_healthcare, 2022, doi:10.3390/healthcare10112326_

Round 1

Reviewer 1 Report

In your study, you researched the psychological problems of Mainland Chinese living in Canada during the Covid-19 period. Your work explores an important issue, but some important issues in the article must be corrected or some additions must be made. I tried to write them one by one below.

- First of all, the weakest part of your work is that it is a single scale. Why didn't you think about this when you were constructing the research? In this way, the research and findings part of your study in this journal will not be enough. In other words, you would start with a model that you can establish a mediation relationship with.

-The introductory sentences of your summary part are very good. In the method part, the scales are not written enough. For example, how did you measure your fear of Covid-19? What analyzes did you use and how? Move the analysis information you used in Result to the method section.

- There should be a research question in the introduction part of the study. It should be about which question this research seeks to answer.

- What theory is your work based on? You should write the literature according to this theory.

- You used only one scale which is not enough for a relational study. In addition, you could have used at least two more scales, and there could have been more advanced analysis.

  - I could not see any hypothesis in the study. What assumptions did you do your research on?

- Your literature section should be more detailed. In fact, literature should be written in the content that will underpin each hypothesis you will write.

- Not enough resources were used in the discussion part of the study. The results should be discussed with the literature. There is, but it's not enough.

- Include the implementations for policymakers and researchers by writing the conclusion separately.

Author Response

  1. The weakest part of your work is that it is a single scale. Why didn't you think about this when you were constructing the research? In this way, the research and findings part of your study in this journal will not be enough. In other words, you would start with a model that you can establish a mediation relationship with.

Response: Thank you for this comment. We agree that it would be ideal to include more standardized scales as outcome measures for psychological distress. However, this work was funded through a rapid response grant at the very early stages of the pandemic and required quick action to collect a broad range of data in order to have a preliminary understanding of a variety of responses (e.g., attitudes, behavioral measures, risk predictions, and psychological responses) towards the pandemic within a very tight timeline. We also considered limiting the length of the survey to ensure a high response rate. We think this brief 7-item scale well served this purpose: 1) it broadly covers most typical emotions specific to the pandemic; 2) it is brief and time efficient; 3) it covers non-pathological psychological distress commonly experienced during a pandemic.  

With that said, we should point out that in a subsequently administered follow-up in-depth survey, we did include this scale with some standardized mental health measures, including the DASS-21 and CPDI. The validation analysis showed a consistent high correlation between this scale and other mental health scales. This, however, is beyond the scope of this current report. For details of this follow-up study, please refer to the following publications:

Yu, L., Lecompte, M.; Zhang, W.; Wang, P.; & Yang, L. (2022). Sociodemographic and COVID-related predictors for mental health condition of Mainland Chinese in Canada amidst the pandemic. International Journal of Environmental Research and Public Health, 19(1), 171; https://doi.org/10.3390/ijerph19010171.

Lecompte, M., Counsell, A., & Yang, L. (2022). Demographic predictors of COVID-19 risk perception among Chinese residents in Canada. International Journal of Environmental Research and Public Health, 19, 14448. https://doi.org/10.3390/ ijerph192114448.

  1. The introductory sentences of your summary part are very good. In the method part, the scales are not written enough. For example, how did you measure your fear of Covid-19? What analyzes did you use and how? Move the analysis information you used in Result to the method section.

Response: We added more clarifications to describe how each item of the scale was worded and scored. (p. 4). We are unsure about which information in Results that you want to be moved to the Method section. We did have a “data analysis” section at the end of the method section. We would be happy to adjust if further specific instructions are provided. Thanks!

  1. There should be a research question in the introduction part of the study. It should be about which question this research seeks to answer.

Response: We added the research question at the end of the introduction: “whether higher psychological distress of Mainland Chinese in Canada at the pandemic initial outbreak was predicated by their sociodemographic profiles (e.g., age, gender, education, financial/health condition, employment status, and discrimination)? “ (p. 3).

  1. What theory is your work based on? You should write the literature according to this theory.

Response: As this is a rapid response survey to quickly collect information to have a broad understanding of our responses to the pandemic, it is largely an explorative study. There is not a specific guiding theory. Nevertheless, we added the double jeopardy hypothesis in the introduction (p. 2) and the discussion (p. 8) sections as a theoretical perspective that motivated the study and offered an interpretation for the results. We also identified some practical implications of this study in the introduction to highlight the importance of the study: “This is important given the time-sensitive behaviour measures implemented at that time. The results will shed light on future public health policy and measures to best mitigate psychological distress or mental health issues in a timely manner at the initial pandemic outbreak“ (p. 3).

  1. You used only one scale which is not enough for a relational study. In addition, you could have used at least two more scales, and there could have been more advanced analysis.

Response: We admit that this is a limitation of this study and recognized it as a limitation in the discussion section. But given the rapid response nature of this work, we think this brief 7-item survey well serves our specific purpose of this study. In a subsequent follow-up survey, we did include this scale with other standardized mental health and psychological distress measures. Results showed a consistent positive correlation between the scales, confirming the validity of this scale to capture overall non-pathological psychological distress during a pandemic. Please see our detailed response to your first comment above.

  1. I could not see any hypothesis in the study. What assumptions did you do your research on?

Response: We added specific hypothesis at the end of the introduction: “it was hypothesized that higher psychological distress during the pandemic would be predicted by such sociodemographic variables as younger age, being a woman, poor health/financial status, lower education, unstable employment, and perceived or experienced anti-Chinese discrimination” (p. 3).

  1. Your literature section should be more detailed. In fact, literature should be written in the content that will underpin each hypothesis you will write.

Response: Given that this study is an explorative study to rapidly assess a broad range of responses towards the pandemic at the initial pandemic outcome stage, there isn’t a specific guiding theoretical foundation for this study. Nevertheless, we further streamlined and tightened up the literature review to be closely consistent with the main question and hypothesis of the study (p 1-3). We also added the double jeopardy hypothesis in the introduction (p. 2) and the discussion (p. 8) sections as a theoretical perspective that motivated the study and offered an interpretation for the results.

  1. Not enough resources were used in the discussion part of the study. The results should be discussed with the literature. There is, but it's not enough.

Response: To address this comment, we added more specific discussion in light of the literature (p. 8).

  1. Include the implementations for policymakers and researchers by writing the conclusion separately.

Response: Following this suggestion, we added the conclusion section at the end (p. 9). In this section, we also included implications for future research direction and practical implication in public health policy.

Author Response

  1. The Introduction has been divided into 4 subsections what made the text clearer. However,

this paragraph is not complete. It is the weakest part of this manuscript. It must be improved

because it lacks of:

- hypotheses,

- the main aim of the work,

- main conclusions.

Response: Following this suggestion, we identified the main aim of this study: “As a rapid response to the pandemic outbreak, the current study aimed to fill this gap to have a quick assessment of non-pathological psychological distress towards COVID-19 pandemic and identify the related sociodemographic predictors among Mainland Chinese in Canada at the early stage of the pandemic” (p. 3), specified the main research question and the hypothesis at the end of the introduction (p. 3, also see our responses to the Item 3 and 5 of Reviewer 1). We also added a separate conclusion section at the end of the discussion section (p. 9).

  1. As we read in the Instructions for Authors: The introduction should briefly place the study in a

broad context and highlight why it is important. It should define the purpose of the work and its

significance, including specific hypotheses being tested. The current state of the research field should be

reviewed carefully and key publications cited. Please highlight controversial and diverging hypotheses

when necessary. Finally, briefly mention the main aim of the work and highlight the main conclusions. Keep the introduction comprehensible to scientists working outside the topic of the paper

Response: Following this suggestion, we streamlined the literature review to be consistent with the research question. We highlighted the importance of the study as “This is important given the time-sensitive behaviour measures implemented at that time. The results will shed light on future public health policy and measures to best mitigate psychological distress or mental health issues in a timely manner at the initial pandemic outbreak” (p. 3).

We also clearly identified the specific aim, question, and the related hypothesis at the end of the introduction (p. 3. please also see our responses to the Item 3 and 6 of Reviewer 1 and the Item 1 of Reviewer 2 for detailed responses). We also proofread closely to ensure the writing is easy to follow.

  1. The subsection 1.4. Present Study should be changed to 1.4. The aim of the Study and be fitted to Instructions for Authors. Information in lines 110-115 is not a part of this paragraph.

Response: We changed the heading accordingly “1.4 The Aim of the Current Study”. We removed lines 110-115 as similar information could be found in the methods section.

  1. Materials and methods

Lines 118-120. How many respondents did you get via:

- WeChat,

- other online platforms such as emails and websites (what websites ? educational institutions,

administration institutions?).

Response: For efficiency purposes, we distributed our survey simultaneously through WeChat (private and group chat forums and moment posts), emails to collaborative Chinese community organizations, participating lab websites, and the Centre for New Immigrant Well-being (CNIW) website. We clarified this information on p. 3. Given all the responses are anonymous, however, we are unable to track which respondent is recruited from which channel.

  1. In my opinion, using a social media platform is a clear limitation of the study. Not mentioned

in the proper section.

Response: We identified this as a limitation in our discussion section (p. 9).

  1. Results: Please correct the title of this section in the plural.

Response: Corrected, thank you.

  1. References: The references do not meet the requirements of the Journal. Look at the Instructions for

Authors: “in the text, reference numbers should be placed in square brackets [], and placed before the punctuation; for example [1], [1 3] or [1,3]. For embedded citations in the text with pagination, use both parentheses and brackets to indicate the reference number and page numbers; for example [5] (p. 10). or [6] (pp.101 https://www.mdpi.com/journal/healthcare/instructions , accessed on 23.10.2022).

Response: we closely reviewed all the citations and references to ensure they match this required format.

Round 2

Reviewer 1 Report

You wrote the resarch question in your study, but you need to evaluate that question at the end of the discussion part according to your research results. You should write at least one paragraph.

I couldn't see any implementations under the heading of limitations and implementations. It is written under conclusion part but not enough.

You can not do more research and analyses than this level using one scale.

You have English spelling problems in your manuscript.

Author Response

1. You wrote the research question in your study, but you need to evaluate that question at the end of the discussion part according to your research results. You should write at least one paragraph.

RESPONSE: We added a paragraph at the beginning of the discussion section to refresh the research question and summarize main results (p. 8). We also added a paragraph at the end to evaluate the research questions in light of the results (p. 9)

2. I couldn't see any implementations under the heading of limitations and implementations. It is written under conclusion part but not enough.

RESPONSE: Thank you for catching this. We removed “implication” from this heading and streamlined a bit on the implications of the study results in the conclusion section (p. 10).

3. You can not do more research and analyses than this level using one scale. You have English spelling problems in your manuscript.

RESPONSE: We acknowledged the “single scale” as a limitation of this study and checked throughout the manuscript for typographic and spelling errors.